# Once-Weekly Subcutaneous Semaglutide Improves Fatty Liver Disease in Patients with Type 2 Diabetes: A 52-Week Prospective Real-Life Study

**DOI:** 10.3390/nu14214673

**Published:** 2022-11-04

**Authors:** Sara Volpe, Giuseppe Lisco, Margherita Fanelli, Davide Racaniello, Valentina Colaianni, Domenico Triggiani, Rossella Donghia, Lucilla Crudele, Roberta Rinaldi, Carlo Sabbà, Vincenzo Triggiani, Giovanni De Pergola, Giuseppina Piazzolla

**Affiliations:** 1Interdisciplinary Department of Medicine, School of Medicine, University of Bari “Aldo Moro”, Piazza Giulio Cesare 11, 70124 Bari, Italy; 2National Institute of Gastroenterology, “Saverio de Bellis”, Research Hospital, Castellana Grotte, Via Turi 27, 70013 Bari, Italy

**Keywords:** nonalcoholic fatty liver disease (NAFLD), liver steatosis, metabolic associated fatty liver disease (MAFLD), type 2 diabetes, obesity, semaglutide, prospective study, real-life study

## Abstract

Background. Nonalcoholic fatty liver disease (NAFLD) is commonly observed in patients with type 2 diabetes (T2D). Semaglutide, a glucagon-like peptide 1 receptor agonist, may have a therapeutic role by targeting common mechanisms involved in the pathophysiology of T2D and NAFLD. The study aimed to assess the effectiveness of Semaglutide on NAFLD in patients with T2D. Methods. Forty-eight patients were treated with subcutaneous Semaglutide in add-on to metformin for 52 weeks. After the baseline visit (T0), follow-up was scheduled quarterly (T3, and T6) and then at 12 months of therapy (T12). During each visit, body composition was analyzed by phase-sensitive bio-impedance, and NAFLD was diagnosed and staged by Ultrasound (US) imaging. Surrogate biomarkers of NAFLD were also calculated and followed over time. Results. A significant decrease in anthropometric and glucometabolic parameters, insulin resistance, liver enzymes, and laboratory indices of hepatic steatosis was observed during treatment. Similarly, fat mass and visceral adipose tissue (VAT) decreased over time more than skeletal muscle and free-fat mass. US-assessed VAT thickness and the 12-point steatosis score also declined at T3 up to T12. Liver steatosis improved in most patients (70%), showing a reduction by at least one class in the semiquantitative US staging. Conclusion. Besides glucose control and body composition improvements, Semaglutide was effective in ameliorating the clinical appearance and severity of NAFLD in T2D patients.

## 1. Background

Nonalcoholic fatty liver disease (NAFLD) is the foremost liver disease in the world, affecting up to 30% of the general population [1]. Its prevalence rises to more than 70% among individuals with metabolic disorders such as metabolic syndrome, type 2 diabetes (T2D), and obesity [2,3].

NAFLD is the result of a metabolic disorder related to a sedentary lifestyle, excessive calorie intake, insulin resistance (IR) [4], as well as genetic and environmental factors [5]. A positive family history, visceral obesity, diabetes mellitus and insulin resistance, and high levels of triglycerides with low circulating High-Density Lipoproteins (HDL) are the most relevant factors associated with NAFLD [6,7,8].

The disease is characterized by macrovesicular fat storage in more than 5% of hepatocytes in non-drinkers or mild drinkers (ethanol consumption <30 g/day in men and <20 g/day in women) [9]. A dysfunctional intrahepatic lipid metabolism is the key element in patients diagnosed with NAFLD, who risk a possible progression to nonalcoholic steatohepatitis (NASH), liver cirrhosis, and, ultimately, hepatocellular carcinoma [10].

The association between NAFLD and metabolic disorders is strong, with a bidirectional relationship. Insulin resistance induces an imbalance between lipid production and disposal, leading to lipid accumulation in hepatocytes, chronic low-grade inflammation, progression to NASH, and adverse hepatic and extrahepatic outcomes [11]. T2D and its related comorbidities, including visceral obesity, arterial hypertension, and dyslipidemia, are likely to accelerate the progression of liver disease from NAFLD to NASH and liver cirrhosis. On the other hand, NAFLD may deteriorate hepatic insulin sensitivity in a self-renewing vicious cycle that further worsens glucometabolic control [12].

To reinforce the association between T2D and NAFLD, a recently recognized definition of hepatic steatosis in patients with such metabolic disorders has been devised, namely Metabolic Associated Fatty Liver Disease (MAFLD) [13]. Compared to NAFLD, MAFLD seems to be associated with a higher risk of disease progression, and so should be recognized and treated early to avoid harmful consequences over time [14]. In addition, patients with MAFLD are at higher risk of mortality due to cardiovascular (CV) and renal diseases, sleep apnea, hepatic and extrahepatic malignancies such as colorectal and breast cancer [15].

The assessment of body composition is a promising approach for studying the pathogenesis and prognosis of NAFLD in T2D patients [16]. Low skeletal muscle mass (SMM), clinically known as sarcopenia, is associated with a greater risk of NAFLD and fibrosis [17]. In addition, the prevalence of NAFLD has recently been shown to be significantly higher in T2D patients with sarcopenia. With particular reference to this point, it should be emphasized that the simultaneous loss of SMM and accumulation of visceral adipose tissue (VAT), the so-named sarcopenic visceral obesity, is associated with higher levels of IR and metabolic impairment, increasing the risk of NAFLD [18,19].

A healthy diet, lifestyle changes, and physical activity are the first-line therapy of NAFLD, demonstrated to improve IR, reduce systemic inflammation, promote weight loss, reduce fat accumulation, increase SMM, and, lastly, prevent negative outcomes [20]. However, long-term compliance and adherence to lifestyle modifications are usually limited and only a few patients achieve and maintain therapeutic goals. On the other hand, anti-fibrotic treatments should only be considered in patients with moderate-to-severe NASH, or those at high risk of disease progression [21]. The number of candidates for this cure is too scanty, so that most patients with NAFLD usually receive poor or non-specific treatments.

Thiazolidinediones, glucagon-like peptide-1 receptor agonists (GLP-1 RAs), and sodium-glucose transporter type 2 inhibitors (SGLT2i) may have an interesting role as pharmacological treatment of MAFLD, since they promote loss of VAT, reduce systemic inflammation, and provide CV protection. In particular, GLP1-RAs and SGLT2i, but not thiazolidinediones, promote weight loss, an important therapeutic target in this setting. GLP-1 RAs appear to have the most promising therapeutic role in NAFLD [22,23], although the indications provided so far are unclear.

Semaglutide is one of the most potent GLP-1 RAs, as demonstrated by its beneficial effects on glucose control [24,25], CV and renal protection [26,27], weight loss, and improving body composition [28]. The pathophysiological background of CV and renal diseases, T2D, obesity, and NAFLD is close, and the beneficial effect of a single treatment on composite endpoints is therefore desirable. To our knowledge, no real-life data assessing the effects of once-weekly Semaglutide on NAFLD in T2D patients have yet been reported.

The aim of this study was to evaluate the effectiveness of diabetes therapeutic dosages of subcutaneous Semaglutide on fatty liver disease and surrogate markers of NAFLD in patients with T2D, treated once weekly for 52 weeks.

## 2. Methods

### 2.1. Study Design, Institution, and Ethics

This is a 52-week prospective, single-arm, real-life study. The study was carried out in the Metabolic Disorders Outpatients Clinic of the Department of Internal Medicine at the University of Bari, Italy, in accordance with the general ethical principles for medical research involving human subjects of the Declaration of Helsinki. The study protocol was formally approved by the Ethics Committee of the University of Bari (n. 6468 version 2, approved on 09/14/2020).

### 2.2. Screening for Eligibility of Study Participants

A total of 150 patients with T2D attending our clinic were screened for eligibility from 1 January to 30 June 2021. The patients were scheduled for intensification of antihyperglycemic therapy due to poor glycemic control, background cardiovascular and renal status, excess weight, or need for weight loss. Among them, 48 (32%) patients (mean age 57.7 ± 8.4 years; men 54.2%) who were willing to receive once-weekly Semaglutide subcutaneously were included in the study. Only 2 of 48 patients discontinued Semaglutide at T6 due to a relevant increase in pancreatic enzymes, and underwent further investigation.

### 2.3. Inclusion Criteria

Inclusion criteria were an established diagnosis of T2D, age > 18 years, stable estimated glomerular filtration rate (eGFR) > 15 mL/min/1.73 m^2^, eligibility for GLP-1 RAs intensification according to current recommendations and guidelines (uncontrolled T2D, i.e., glycated hemoglobin >7% while on oral antihyperglycemic medications or, independently of baseline HbA1c or individualized glycemic targets, in cases of established CV diseases or a high CV risk), and the need to minimize weight gain or promote weight loss.

### 2.4. Exclusion Criteria

These included other forms of diabetes mellitus, pregnancy or lactation, inadequate compliance, or contraindications to GLP-1 RAs, previous or ongoing treatment with pioglitazone and/or SGLT2i and/or GLP-1 RAs, inadequate ability to comply with follow-up or to provide informed consent, assumption of oral contraceptives or corticosteroids, viral hepatitis B and C, and excessive ethanol consumption (more than 30 g/day in men and 20 g/day in women) [9].

### 2.5. Study Protocol

Eligible patients were fully informed about the study’s purposes and gave written consent to participate. Follow-up visits were carried out, during four scheduled appointments, at baseline (T0), and after 3, 6, and 12 months of treatment (T3, T6, and T12). A complete medical history was collected, and a physical examination was carried out during each follow-up visit. The clinical and anthropometric parameters analyzed were office arterial pressure, heart rate, body weight (BW), waist circumference (WC), and body mass index (BMI). Laboratory tests were prescribed according to the general principles of clinical practice and the recommended follow-up. In particular, the laboratory tests included a complete total blood count, fasting glycemia, glycated hemoglobin (HbA1c), serum creatinine with eGFR, serum aspartate aminotransferase (AST), alanine aminotransferase (ALT), gamma-glutamyl transferase (γGT), insulin, and C-peptide. Liver ultrasound (US) examination with the grading of hepatic steatosis was also performed.

Once-weekly (qw) Semaglutide was administered subcutaneously in accordance with general recommendations and guidelines. As per good clinical practice, a dose of 0.25 mg qw was prescribed for 4 consecutive weeks and then raised to 0.5 mg qw. After six months of treatment, Semaglutide was further intensified to 1 mg qw to reinforce extra glycemic effects, especially for inducing additional body weight loss, when indicated. All participants were on metformin at the maximum tolerated dose, and Semaglutide was prescribed as an add-on. Glinides, sulfonylureas, dipeptidyl peptidase type IV inhibitors, were discontinued properly. No patient was on insulin therapy at baseline, nor was it prescribed to any patient before the end of the study.

The Homeostasis Model Assessment of Insulin Resistance (HOMA-IR) index was calculated as (fasting insulin × fasting glucose)/405, and used as an indirect measure of systemic insulin resistance (normal range 0.23–2.5).

The AST to Platelet Ratio Index (APRI) score was calculated for estimating the risk of liver fibrosis and cirrhosis according to the following formula: [(AST/upper normal value] × 100]/number of platelets. A value greater than 0.5 was considered highly indicative of an increased risk of liver fibrosis, while a value exceeding 1 was suggestive of an increased risk of cirrhosis [29].

The Hepatic Steatosis Index (HSI) was used to estimate the risk of liver steatosis, and was calculated according to the following formula: [8 × (AST/ALT) + BMI + 2 + 2 if woman]. A value over 36 was indicative of NAFLD and a value lower than 30 should exclude it [30].

The Fatty Liver Index (FLI) was calculated according to the formula: FLI = (e^0.953 × log^_e_
^(triglycerides) + 0.139 × BMI + 0.718 × log^_e_
^(γgt) + 0.053 × waist circumference − 15.745^)/(1 + e^0.953 × log^_e_
^(triglycerides) + 0.139 × BMI + 0.718 × log^_e_
^(γgt) + 0.053 × waist circumference − 15.745^) × 100. An FLI lower than 30 (negative likelihood ratio up to 0.2) suggests that the presence of liver steatosis may be excluded, whereas an FLI equal to or more than 60 (positive likelihood ratio starting from 4.3) was highly indicative of liver steatosis [31].

Body composition was non-invasively measured by phase-sensitive, octopolar Segmental Multifrequency Bioelectrical Impedance Analysis (SMF-BIA; Seca mBCA 525; Seca GmbH & Co., KG, Hamburg, Germany), as described elsewhere [28]. Briefly, the measurements were obtained with patients in a supine position with each leg lying at an angle of 45° and each arm at an angle of 30° from the trunk. To provide accurate measurements, patients fasted for 8 h and rested for at least 8 h. Body impedance was measured using an alternating current at 100 μA with frequencies ranging from 1–500 kHz. The raw data (Rz, Xc) were processed by Seca Analytics 115 software to obtain the following values: total body water (TBW), extracellular water (ECW), SMM, skeletal muscle index (SMI) [32], fat mass index (FMI), fat-free mass index (FFMI), VAT.

The hand grip (HG) strength test was performed using a manual hydraulic dynamometer (Lafayette Instrument, Lafayette, IN, USA) to assess the muscle strength in both hands [33]. Three separate repeat measurements were recorded for each hand and the best value was selected for the analyses. To more accurately assess the muscle quality, the muscle quality index (MQI), which correlates the skeletal muscle mass with the functional data expressed by the HG, was also calculated by dividing the HG strength value by the SMM (kg/kg).

VAT thickness was also estimated with a US method (US-VAT) using a General Electrics Logiq E9 ultrasound machine (GE Healthcare, Milwaukee, WI, USA) according to a validated method [34]. Briefly, the US-VAT was evaluated by placing a convex US probe (1–6 MHz) on the abdominal median line around the umbilical scar to visualize the abdominal aorta before its bifurcation. The distance between the inner line of the recti muscles and the abdominal aorta anterior wall was then measured at the end of expiration. Using the same method, the distance between the outer line of the recti muscle and the skin surface was considered as the thickness of the subcutaneous adipose tissue [34].

Liver steatosis was graded on the basis of US features assessed with a General Electrics Logiq E9 ultrasound machine. First of all, according to a validated semiquantitative score, NAFLD was classified as absent (0), mild (1), moderate (2), or severe (3) based on standardized US characteristics of liver echogenicity [35]. Moreover, in order to observe minimal variations in steatosis during treatment, the Hamaguchi 6-point score [36] was modified to obtain a more detailed ultrasound liver steatosis score (US-LSS), ranging from 0 to 12 points (Table 1). More precisely, four typical US characteristics were considered for grading the severity of liver steatosis: liver brightness, US appearance of intrahepatic vessels, parenchymal echotexture, and US beam attenuation. From zero to three points were assigned to each feature (Table 1 and Figure 1).

### 2.6. Study Outcomes

The primary study outcome was to estimate the mean changes in serum liver enzymes (AST, ALT, and γGT), validated clinical scores (APRI, HSI, and FLI), and US liver appearance over the entire follow-up (T0, T3, T6, and T12).

Additional study outcomes included mean changes in glucometabolic control (fasting glycemia, HbA1c, lipid profile, and insulin resistance), and body weight and composition (BMI, WC, bio-impedance analysis).

### 2.7. Statistical Analysis

Descriptive statistics were represented as mean, standard deviation, frequency, and percentage. Changes in the variables over time (i.e., T0, T3, T6, and T12) were evaluated by Mixed Models for repeated measures, and means were estimated by the least-squares method. Pearson correlation for the association among quantitative variables was used, while the Kendall coefficient was employed when appropriate, and the chi-square test was used for the association between categorical variables. To evaluate significant changes in steatosis degree from the first to the last observation time, the McNemar Bowker test was used. Comparisons between Responder and Non-Responder patients were evaluated by unpaired t Student. The analyses were carried out on the entire study population and split by gender. Statistical analysis was performed using SAS software 9.4. (SAS Institute Inc., Cary, NC, USA).

## 3. Results

### 3.1. Baseline Characteristics of Study Population (n = 48)

The baseline characteristics of study participants are shown in Table 2. The patients were predominantly men (54.2%), with a male-to-female ratio (1.2:1). The median values of BMI (36.9 kg/m^2^) and WC (120 cm) suggested a high prevalence of central obesity among the study participants, who also exhibited clinically relevant insulin resistance (median HOMA-IR value 5.5; normal range 0.23–2.5). According to baseline BMI, patients were classified as normal weight (BMI < 25 kg/m^2^) only in 2.1% of the cases, overweight (25 < BMI < 30 kg/m^2^) in 6.2%, with mild obesity (30 < BMI < 35 kg/m^2^) in 27.1%, and moderate-to-severe obesity (BMI ≥ 35 kg/m^2^) in 64.6% (Figure 2).

Mean baseline liver enzymes (AST, ALT, γGT) were in the normal range (<34, <49, and <73 U/L, respectively) as indicated in Table 2. According to background cardiovascular risk, 17 of 48 patients did not achieve an appropriate target of Low-Density Lipoprotein (LDL) cholesterol, and anti-hypercholesterolemia therapy was, therefore, intensified at T0.

The APRI score was within the normal range (<0.5) in all. The HSI, FLI, and quantitative LSS suggested the presence of NAFLD in all patients (Table 2). NAFLD was classified by the semiquantitative US-score as mild in 12% of patients, moderate in 44%, and severe in the remaining (Figure 3).

To assess the internal consistency of measurements, some statistical correlations were calculated at baseline (Table 3). In particular, both VAT evaluated by BIA (BIA-VAT) and FMI were positively correlated with HSI, FLI, US-VAT, and LSS. In addition, the HOMA-IR index was found to correlate positively with FLI and LSS, and negatively with MQI (Table 3). Of note, our 12-point LSS correlated significantly with the semiquantitative 4-point score (r = 0.76; *p* < 0.001). 

### 3.2. Effects on Anthropometric and Glucometabolic Parameters

Body weight loss was statistically and clinically relevant at each follow-up visit compared to baseline, with a mean loss in body weight of 7.4% at T3, 9.2% at T6, and 10.3% at T12 (Figure 4). Similar improvements were found for the BMI and Waist Circumference (Figure 4).

After one year of therapy, a significant improvement in fasting glycemia, HbA1c, serum insulin, and HOMA-IR index (Figure 4), as well as in serum lipids (Figure 5) was observed in the study population. More importantly, most of these variables improved significantly after 3 months of therapy and the benefits persisted up to T12, except for fasting serum triglycerides and insulin levels, and the HOMA-IR index which decreased after 6 months up to T12 (Figure 4 and Figure 5). HDL cholesterol was found to decrease slightly after three months of treatment (T3), and then increased progressively up to T12, when it appeared to be significantly higher compared to baseline (Figure 5).

No significant changes were observed throughout the study period in systolic and diastolic blood pressure, heart rate, eGFR, and fasting serum c-peptide (Figure 6).

### 3.3. Liver Enzymes, Scores and US Appearance of NAFLD

As shown in Figure 7, AST, ALT, and γGT decreased significantly during the study period. Similarly, the APRI score, and the biometrical indices of NAFLD (HSI and FLI), US-VAT, and US-LSS (Figure 8) were found to be significantly reduced already at T3 up to T12. The US-subcutaneous fat showed a different trend, as it lowered significantly at T3 and T6, but then it thickened again, so that T12 values did not differ from T0 (Figure 8).

Most patients (70%) achieved a significant improvement in liver steatosis severity from baseline by the end of the observation period, as expressed by at least one-class reduction in the 4-point semiquantitative US staging (*p* < 0.001). These patients were defined as Responders (R) to treatment. In the remaining 30%, namely Non-Responders (NR), the steatosis grading was unchanged; no patient got worse. Importantly, the improvement in liver steatosis was independent of gender (*p* = 0.16) and intensification of anti-hypercholesterolemic therapy at baseline (*p* = 0.11). No differences were found between R and NR with regard to age (*p* = 0.12) and duration of T2D (*p* = 0.19), or changes in BMI (*p* = 0.39), HOMA-IR (*p* = 0.28), and liver enzymes (*p* > 0.05) from baseline to the end of the study.

### 3.4. Body Composition

Table 4 summarizes the variations of the main bio-impedance parameters throughout the study period. The FMI and BIA-VAT decreased significantly at each observation time (T3, T6, and T12) compared to baseline (T0). The FFMI and SMI also declined from baseline to T12 but the magnitude of loss was clinically less relevant. Indeed, at the end of the study, the decrease in SMI exceeded 40% of total weight loss in only 18% of patients and, most importantly, the HG and MQI (both indicative of muscular functional status) were not significantly different at each time compared to baseline (see Table 4). Finally, changes in the SMM/BIA-VAT ratio were analyzed to assess the evolution of sarcopenic visceral obesity during Semaglutide therapy. As shown in Table 4, this ratio progressively increased, reaching significantly higher values than at baseline after one year of therapy. Total and extracellular body water did not change throughout the entire study period (Table 4).

Of note, the changes registered through the follow-up were comparable between men and women.

## 4. Discussion

In this 52-week prospective, real-life study, we evaluated the effect of once-weekly Semaglutide on NAFLD in patients with T2D. Our results showed that Semaglutide significantly reduced NAFLD severity, besides showing beneficial effects on glycemic control, body weight, and composition. It is worth considering that obesity and T2D are the leading risk factors for MAFLD and vice versa. In this context, GLP-1 RAs offer a promising strategy for the pharmacological treatment of MAFLD as they induce considerable weight loss and an insulin-sensitizing effect, and reduce VAT. In particular, they reduce de novo lipogenesis, enhance the mitochondrial beta-oxidation of free fatty acids, reduce systemic, adipose, and hepatic insulin resistance, and increase the clearance of very low-density lipoproteins [37]. Moreover, GLP-1 RAs reduce intrahepatic fat accumulation and hepatic cytolysis, also reducing tissue inflammation [38,39]. Thanks to these mechanisms, it is reasonable to expect that GLP-1 RAs may prevent or halt liver fibrosis and progression to liver cirrhosis.

An accurate estimation of intrahepatic fat accumulation is an important stage in the diagnostic work-up of diabetes-related comorbidities and chronic complications. Non-invasive imaging techniques, such as US, and laboratory surrogate biomarkers of liver steatosis have been validated to diagnose NAFLD [40]. Liver biopsy is no longer the gold standard for assessing the presence of NAFLD and NASH, as it is an invasive procedure with not infrequent complications, difficult to repeat, and providing information on a limited sample of tissue. MRI estimates the proton density fat-fraction in the liver with more relevant sensitivity and accuracy but its routine use in clinical practice poses some challenges due to high costs, especially in cases requiring close follow-up.

In our study, NAFLD was diagnosed and staged over time by US B-mode imaging, an inexpensive and easy-to-perform diagnostic tool that allows subjective examination of the severity of fat infiltration in the liver [35]. US is the first-choice technique in routine clinical practice to assess liver disease, despite some limitations that include: (i) enlarged abdominal wall thickness (i.e., relevant obesity); (ii) low sensitivity, especially in detecting steatosis in <20% of the entire hepatic parenchyma; (iii) low specificity in case of inflammation and fibrosis. Surrogate markers of NAFLD, such as HSI and FLI, have also been calculated with the aim of increasing the accuracy and reliability of NAFLD diagnosis. As a relevant finding, all patients exhibited US signs of NAFLD at baseline and most of them (88%) had moderate to severe steatosis (stages 2 to 3). The finding of a baseline positive correlation between visceral adipose tissue and fat mass index, assessed by the BIA, with US-evaluated visceral fat as well as with all liver steatosis indices and scores, supports the importance of performing a correct assessment of body composition to complete the clinical picture of patients with MAFLD.

Long-term data on the effectiveness of Semaglutide in real-life settings fully met expectations. Besides durable improvements in glucose control, body weight, and composition, Semaglutide increased the levels of serum HDL cholesterol after one year of therapy. Low HDL cholesterol is classically considered a CV risk factor, showing a crucial role of HDL in sustaining reverse cholesterol transport, reducing cholesterol accumulation in the vascular wall, smothering intimal inflammation, and lastly preventing or halting atherosclerosis [41]. As an important issue, low levels of HDL are a classic feature of MAFLD, being a part of the pathophysiological disorder involving lipid metabolism. No pharmacological therapy has been shown to increase HDL cholesterol sufficiently. At least partially, the improvement we observed could be the result of an enhanced intrahepatic synthesis of HDL which, in turn, might have effects on MAFLD improvement. If confirmed by further studies, these data could contribute to a better comprehension of the CV beneficial effects of Semaglutide in patients with T2D and MAFLD.

An improvement in body composition with Semaglutide was demonstrated in our previous study [28], and was herein confirmed after 52 weeks. In particular, fat mass and, more importantly, VAT, which is the leading contributor to insulin resistance and CV risk, significantly and progressively improved through the follow-up and had further decreased by T12 compared to T6. A less evident, but still significant, progressive reduction in lean and skeletal mass was also observed. This reduction was not reflected in impaired skeletal muscle performance as no significant changes in muscle strength and muscle quality index were found up to 52 weeks of treatment. Moreover, in most patients, the weight loss was healthy and favorable since they lost fat rather than muscle mass by more than 40% of the total amount of weight loss. The progressive increase in the SMM/BIA-VAT ratio clearly testifies to the marked improvement in the sarcopenic visceral obesity state of T2D patients following therapy with Semaglutide. The preservation of skeletal muscle mass is a key element in maintaining insulin sensitivity; hence, preserving skeletal muscle mass and performance while decreasing VAT should be considered one therapeutic goal of T2D and MAFLD.

Our findings highlighted a progressive reduction of serum liver enzyme concentrations and an improvement of APRI score from T3 to T12. Although baseline values of both were normal, suggesting that liver inflammation, cytolysis, and fibrosis were generally mild or absent in the study population, our data confirmed the potential role of Semaglutide in preventing progression to NASH in real-life. However, focused trials are needed to confirm these results in a cluster of patients with NASH and T2D. The significant decrease in liver steatosis indices (HSI and FLI) was consistent with the concurrent effects of Semaglutide in reducing body weight, BMI, WC, and triglycerides, all influencing the calculation of NAFLD biometric markers. On the other hand, the clinical improvement of liver steatosis was also highlighted by the early decrease in US-assessed liver fat, which was significant soon after 3 months of therapy, and persisted up to 52 weeks. Moreover, the percentage of R patients, exhibiting at least a one-class improvement of liver steatosis severity according to the US-based semiquantitative score, was very high (80%) therefore suggesting that Semaglutide may have a relevant role in treating MAFLD in T2D. There is still debate as to whether GLP-1 RAs may improve liver steatosis directly, since the GLP-1 receptor is not expressed at appreciable levels in the liver [22]. On this basis, one can say that GLP-1 RAs could induce liver steatosis improvement with indirect effects. However, no specific factors that explained a different response to Semaglutide in terms of improvement of liver steatosis (R and NR patients), such as gender, changes in body weight, BMI, WC, and insulin resistance, were identified in the current study. This could be an important issue requiring specific investigation in a broader clinical trial, also with the aim of clarifying whether the presence of NAFLD or NASH can differently affect the response of patients to Semaglutide.

A special consideration should be made about metformin therapy. All participants were on metformin before the study entry and received a stable dose over the follow-up. Although metformin has shown anti-inflammatory properties and potentially positive effects on hepatic steatosis and steatohepatitis, by suppressing de novo lipogenesis, enhancing fatty acid oxidation, and promoting degradation of intracellular lipids that accumulate in hepatocytes [22,42] no significant improvement in US and histological features of NAFLD has been reported in patients treated with metformin, regardless of the concomitant presence of diabetes [22,42]. Semaglutide shares with metformin only two mechanisms of action, that include attenuation of the hepatic glucose production rate and increased peripheral insulin uptake. The two mechanisms are expected to be enhanced when combining metformin with Semaglutide, with a further improvement in fasting glycemia and weight loss, but the contribution of both mechanisms to improving hepatic steatosis, inflammation, and, lastly, fibrosis is probably negligible, although it cannot be ruled out.

In line with our results, two large clinical trials, in which NAFLD was an established comorbidity among the enrolled patients, have recently shown a dose-dependent reduction of serum ALT and high-sensitivity C-reactive protein values after treatment with Semaglutide [38]. In a specific phase II trial in patients with NASH, Semaglutide was administered once daily at different subcutaneous doses (0.1, 0.2, and 0.4 mg) for 72 consecutive weeks, showing promising effects on intrahepatic fat accumulation but without significant changes in terms of liver fibrosis compared to placebo [43]. Similarly, satisfactory effects on liver steatosis but not liver stiffness were confirmed in a study using magnetic resonance imaging to assess changes in liver parenchyma [44]. Finally, a very recent pilot study tested the oral formulation of Semaglutide for 24 weeks in 16 TD2 patients with NAFLD, showing results consistent with our findings [45].

Our study has some strengths and limitations. Strengths include the robust study design, a well-standardized and comprehensive assessment of participants, a long-term follow-up (52 weeks), and the real-life nature of the observations which, in turn, may provide useful information for clinical practice. The main limitations of the study are the small (even if statistically adequate) sample size and the lack of more sensitive, specific, and accurate instrumental examinations providing a quantitative classification of hepatic fat accumulation, such as liver biopsy or magnetic resonance imaging. Based upon the latter point, minor steatosis or small changes over time in hepatic fat deposits would preferably be studied with the Controlled Attenuation Parameter (CAP) algorithm implemented in the transient elastography (FibroScan) system, and the Ultrasound-Guided Attenuation Parameter (UGAP). These imaging techniques are an accurate diagnostic tool for the point-of-care investigation of MAFLD [46], but they are not commonly available in internal medicine clinics.

## 5. Conclusions

Our data show that Semaglutide improves liver steatosis in patients with T2D. Since NAFLD plays a crucial role in the pathophysiology of T2D and related cardiovascular and renal complications, Semaglutide should be considered as an optimal approach for the treatment of patients with T2D and fatty liver disease.

## Figures and Tables

**Figure 1 nutrients-14-04673-f001:**
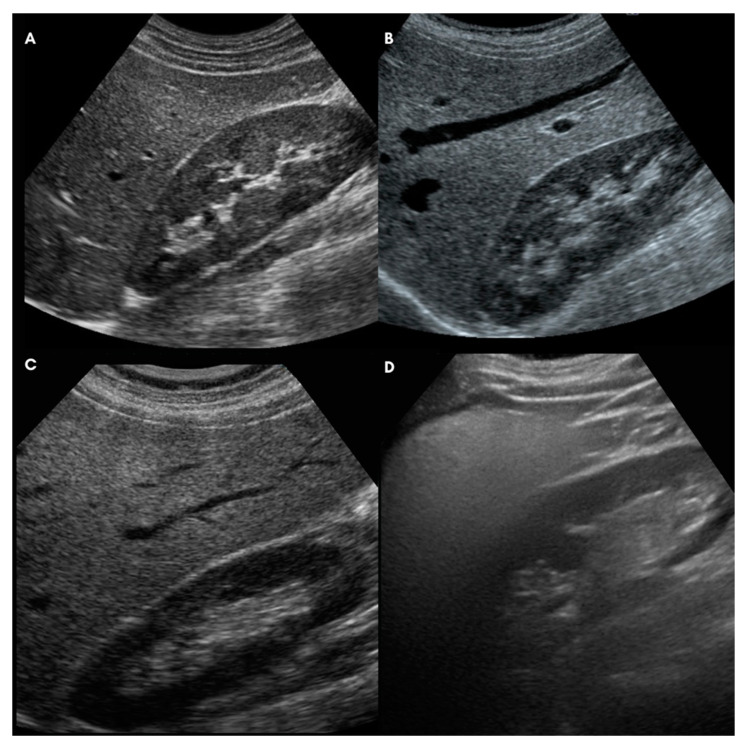
Ultrasound grading of hepatic steatosis: (**A**) Normal Liver (score 0); (**B**) Mild Steatosis (score 1–4); (**C**) Moderate Steatosis (score 5–8); (**D**) Severe Steatosis (score 9–12).

**Figure 2 nutrients-14-04673-f002:**
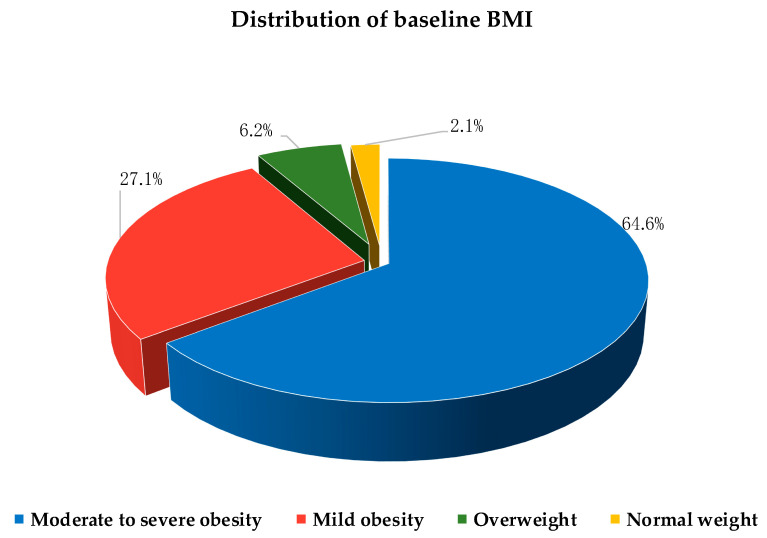
Baseline distribution of BMI classes of the study population: normal weight (BMI < 25 kg/m^2^); overweight (25 < BMI < 30 kg/m^2^); mild obesity (30 < BMI < 35 kg/m^2^); moderate-to-severe obesity (BMI ≥ 35 kg/m^2^).

**Figure 3 nutrients-14-04673-f003:**
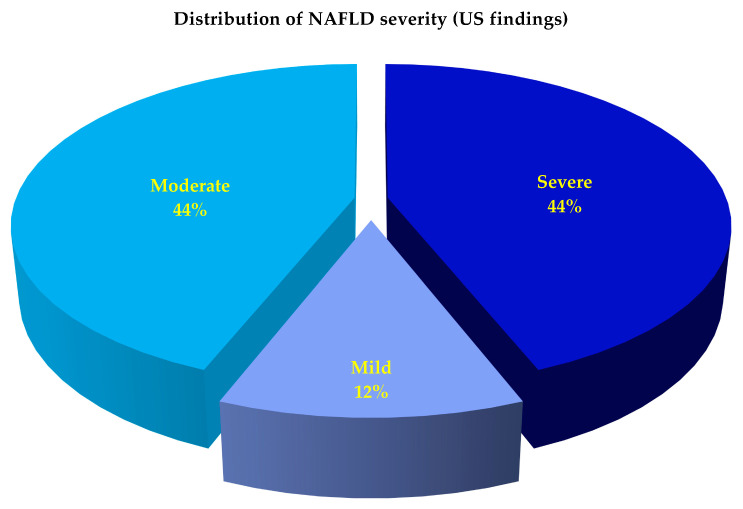
Baseline distribution of NAFLD severity according to a semiquantitative US-score in the study population. Abbreviation: NAFLD = non-alcoholic fatty liver disease; US = ultrasonographic.

**Figure 4 nutrients-14-04673-f004:**
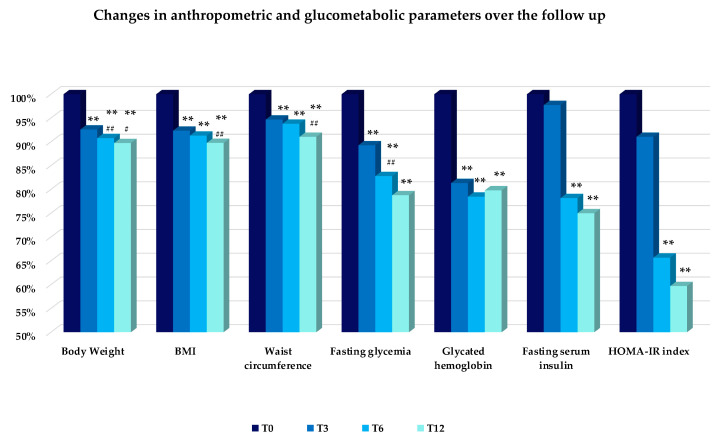
Values are expressed as a percentage of changes in anthropometric (Body Weight, BMI, Waist Circumference), and glucometabolic (Fasting Glycemia, Glycated Hemoglobin, Fasting Serum Insulin, and HOMA-IR index) parameters compared to the baseline (T0 = 100%). Changes versus T0: ** *p* < 0.01. Changes versus former time: # *p* < 0.05; ## *p* < 0.01. Abbreviations: BMI = Body Mass Index; HOMA-IR = homeostasis model assessment of insulin resistance.

**Figure 5 nutrients-14-04673-f005:**
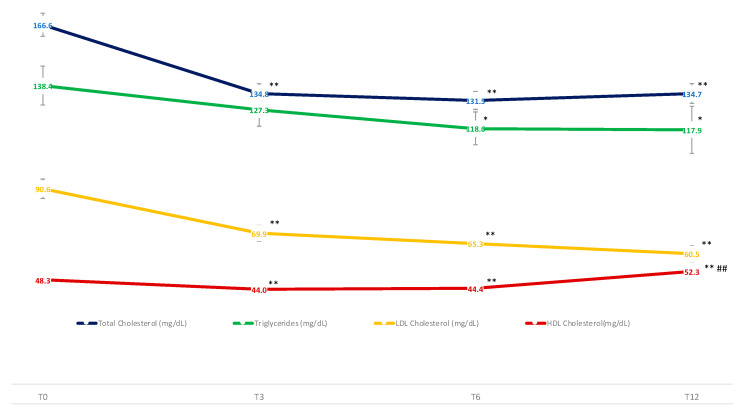
Mean changes in serum values of Total Cholesterol, Low-Density Lipoprotein (LDL) Cholesterol, High-Density Lipoprotein (HDL) Cholesterol, and Triglycerides (mg/dL) through the study period. Means are reported as Least Squared Means estimated by Mixed Model analysis. Variation versus T0: * *p* < 0.05; ** *p* < 0.01. Variation versus previous time: ## *p* < 0.01.

**Figure 6 nutrients-14-04673-f006:**
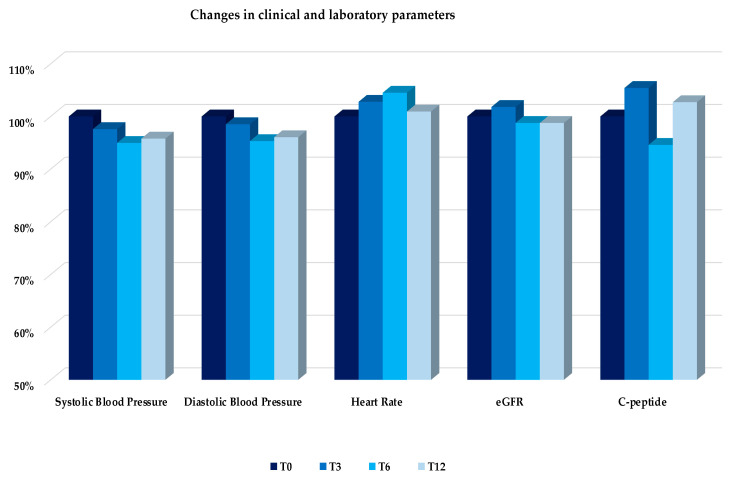
Mean changes, expressed as percentage changes from T0, in clinical parameters, eGFR, and serum C-peptide over the follow-up (T0, T3, T6, T12). No statistically significant changes have been found. Abbreviation: Glomerular filtration rate (eGFR).

**Figure 7 nutrients-14-04673-f007:**
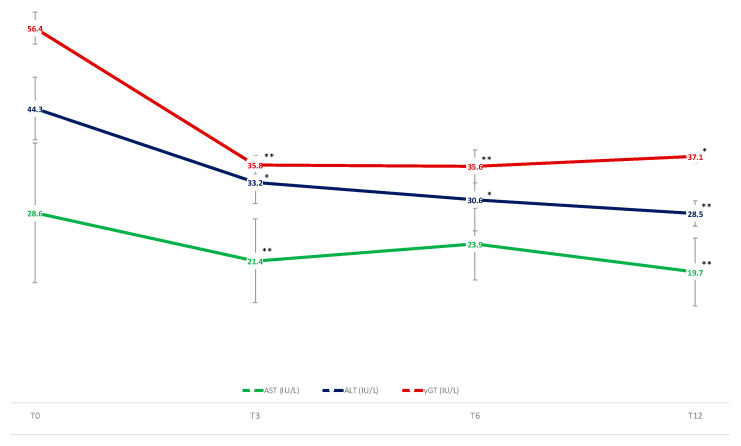
Mean changes in serum values of AST, ALT, and γGT (U/L) over the study period. Means are reported as least squared means estimated by mixed model analysis. Variation versus T0: * *p* < 0.05; ** *p* < 0.01. Abbreviations AST: ASpartate aminoTransferase; ALT: ALanine aminoTransferase; γGT: gamma-glutamyl transferase.

**Figure 8 nutrients-14-04673-f008:**
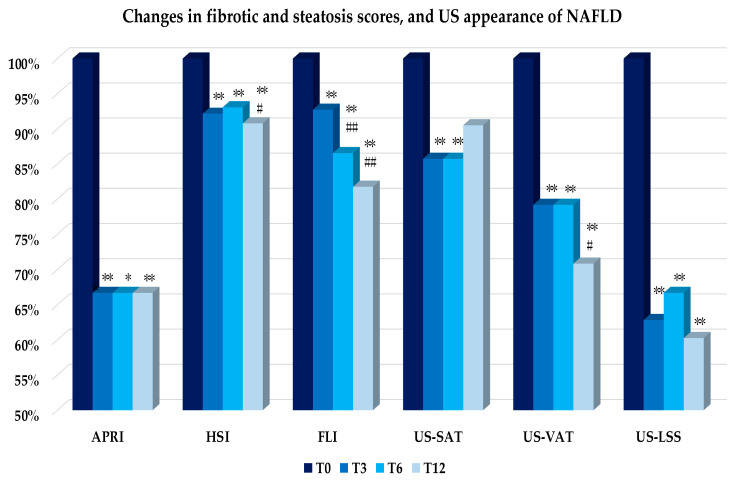
Mean changes, expressed as percentage changes from T0, in biometrical indices of hepatic fibrosis (APRI) and steatosis (HSI, FLI), and US parameters (US-SAT, VAT, and LSS) over the follow-up (T0, T3, T6, T12). Variation versus T0: * *p* < 0.05; ** *p* < 0.01. Variation versus previous time: # *p* < 0.05; ## *p* < 0.01. Abbreviations: APRI: Aspartate aminotransferase to Platelet Ratio Index; HSI: Hepatic Steatosis Index; FLI: Fatty Liver Index; US: ultrasound; SAT: Subcutaneous Adipose Tissue; VAT: Visceral Adipose Tissue; LSS: Liver Steatosis Score.

**Table 1 nutrients-14-04673-t001:** Ultrasound Liver Steatosis Score (US-LSS).

Ultrasound-Liver Steatosis Score (modified from Hamaguchi et al. 2007) [36]
A. Bright liver and hepatorenal echo contrast
0: Bright liver and hepatorenal US contrast are negative
1: Bright liver and/or hepatorenal US contrast are slightly positive
2: Bright liver and/or hepatorenal US contrast are mildly positive
3: Bright liver and/or hepatorenal US contrast are strongly positive
B. Deep attenuation
0: Deep attenuation is completely negative
1: Deep attenuation is mild (the deep parts of the liver are visualized but appear hypoechoic)
2: Deep attenuation is severe (the diaphragm, but not the deep liver parenchyma, can be distinguished)
3: Total barrage of the ultrasound beam (neither deep liver parenchyma nor diaphragm can be distinguished)
C: Vessel blurring
0: Vessel blurring is negative
1: The borders of intrahepatic vessels are not perfectly defined
2: The border of intrahepatic vessels is mildly unclear or the lumen of intrahepatic vessels is narrowed
3: Intrahepatic vessels are not visible (“featureless liver”)
D: Parenchymal US-pattern
0: Parenchymal US-pattern is homogeneous
1: Parenchymal US-pattern is slightly inhomogeneous
2: Parenchymal US-pattern is mildly inhomogeneous
3: Parenchymal US-pattern is severely inhomogeneous.
Ultrasound Steatosis Score (0–12) = A + B + C + D

US-LSS (0–12) was calculated by summing up the single score (from zero to three points) assigned to each US feature: liver brightness (A), US beam attenuation (B), US appearance of intrahepatic vessels (C), parenchymal echotexture (D).

**Table 2 nutrients-14-04673-t002:** Baseline characteristics of the study population (n = 48).

Baseline Characteristics	Mean (S.D.)	Median (Min; Max)
Age (years)	57.7 (8.4)	59 (42;73)
Diabetes duration (years)	6 (6)	3.5 (0; 20)
Body weight (kg)	104.7 (22.5)	99 (70.5; 170)
Body mass index (kg/m^2^)	38.8 (8.3)	36.9 (24.7; 61)
Waist circumference (cm)	123.2 (15.3)	120 (97; 161)
Fasting glycemia (mg/dL)	129.5 (34.7)	122 (87; 231)
Glycated hemoglobin (mmol/mol)	53.2 (20.2)	44.5 (33; 128)
Fasting serum C-peptide (ng/mL)	3.7 (1.5)	3.5 (0.3; 8.1)
Fasting serum insulin (mUI/L)	22.6 (15.4)	19.9 (0.4; 75)
HOMA-IR index	6.8 (4.6)	5.5 (0.1; 21.8)
Total Cholesterol (mg/dL)	166.5 (37.8)	165 (102; 247)
LDL Cholesterol (mg/dL)	90.3 (31.8)	90 (36; 164)
HDL Cholesterol (mg/dL)	47.9 (13.9)	46 (30; 116)
Triglycerides (mg/dL)	140.7 (63.3)	144 (44; 313)
Serum creatinine (mg/dL)	0.9 (0.2)	0.85 (0.40; 1.65)
Glomerular filtration rate (mL/min/1.73 m^2^)	88.9 (21.6)	89.5 (41; 153)
AST (IU/L)	28.2 (16.7)	23 (10; 90)
ALT (IU/L)	43.7 (32.6)	36 (15; 156)
γGT (IU/L)	58.2 (75.6)	39 (12; 430)
APRI score	0.3 (0.2)	0.3 (0.1; 1)
HIS	47.9 (8.8)	45.7 (28.5; 69.6)
FLI	91.7 (7.7)	93.6 (73.4; 100)
US-Subcutaneous Adipose Tissue (cm)	2.2 (0.8)	2 (0.9; 4)
US-Visceral Adipose Tissue (cm)	7.2 (2.8)	6.6 (3; 15.7)
US-LSS	7.8 (2.5)	8 (2; 12)

Descriptive statistics are presented as means, standard deviation, and median, minimum, and maximum values. Abbreviations: HOMA-IR: homeostasis model assessment of insulin resistance; LDL: Low-Density Lipoprotein; HDL: High-Density Lipoprotein; AST: ASpartate aminoTransferase; ALT: ALanine aminoTransferase; γGT: gamma-glutamyl transferase; APRI: Aspartate aminotransferase to Platelet Ratio Index; HSI: Hepatic Steatosis Index; FLI: Fatty Liver Index; US: ultrasound; US-LSS: ultrasound liver steatosis score.

**Table 3 nutrients-14-04673-t003:** Correlations between the examined variables.

BMI	WC	BIA-VAT	US-VAT	APRI	HSI	FLI	US-LSS	US-SAT	FMI	FFMI	SMI	HG	MQI	HOMA IR
**BMI**	0.92	0.71	0.65	0.01	0.96	0.72	0.46	0.31	0.88	0.73	0.56	−0.18	−0.23	0.22
	**	**	**	n.s.	**	**	*	n.s.	**	**	**	n.s.	n.s.	n.s.
	**WC**	0.75	0.71	0.06	0.87	0.79	0.56	0.36	0.79	0.69	0.51	−0.04	−0.17	0.30
		**	**	n.s.	**	**	**	*	**	**	**	n.s.	n.s.	*
		**BIA-VAT**	0.74	0.16	0.59	0.53	0.44	0.13	0.50	0.82	0.69	0.28	−0.21	0.18
			**	n.s.	**	**	**	n.s.	**	**	**	n.s.	n.s.	n.s.
			**US-VAT**	0.01	0.50	0.59	0.57	0.22	0.45	0.62	0.51	0.15	−0.16	0.31
				n.s.	**	**	**	n.s.	**	**	**	n.s.	n.s.	n.s.
				**APRI**	0.02	0.09	−0.05	0.16	−0.05	0.14	0.09	0.10	−0.09	0.16
					n.s.	n.s.	n.s.	n.s.	n.s.	n.s.	n.s.	n.s.	n.s.	n.s.
					**HSI**	0.65	0.35	0.34	0.88	0.62	0.44	−0.30	−0.19	0.16
						**	*	*	**	**	**	n.s.	n.s.	n.s.
						**FLI**	0.53	0.44	0.57	0.51	0.43	−0.10	−0.27	0.41
							**	**	**	**	**	n.s.	n.s.	**
							**US-LSS**	0.37	0.32	0.41	0.35	0.17	−0.17	0.41
								*	*	**	*	n.s.	n.s.	**
								**US-SAT**	0.38	0.05	0.18	−0.11	−0.25	0.48
									*	n.s.	n.s.	n.s.	n.s.	**
									**FMI**	0.41	0.29	−0.25	−0.17	0.12
										**	n.s.	n.s.	n.s.	n.s.
										**FFMI**	0.84	0.14	−0.26	0.21
											**	n.s.	n.s.	n.s.
											**SMI**	0.11	−0.54	0.33
												n.s.	**	*
												**HG**	0.65	0.11
													**	n.s.
													**MQI**	−0.31
														*
														**HOMA IR**


Correlation coefficients and statistical significance are reported at each intersection between the variables. Graduations of green were used for positive correlations, and yellow for negative ones, from lowest to highest correlation coefficients. * *p* < 0.05; ** *p* < 0.01. Abbreviations: BMI: Body Mass Index; WC: Waist Circumference; BIA: Bioelectrical Impedance Analysis; VAT: Visceral Adipose Tissue; US: ultrasound; APRI: Aspartate aminotransferase to Platelet Ratio Index; HSI: Hepatic Steatosis Index; FLI: Fatty Liver Index; LSS: Liver Steatosis Score; SAT: Subcutaneous Adipose Tissue; FMI: Fat Mass Index; HG: Hand-Grip; MQI: Muscle Quality Index; HOMA-IR: homeostasis model assessment of insulin resistance.

**Table 4 nutrients-14-04673-t004:** Mean changes in bio-impedance parameters describing body composition of study participants.

Parameters	Variation over Time			
	T0	T3	T6	T12
Visceral Adipose Tissue (L)	6.2 (0.5)	−0.8 (0.2) **	−1.1 (0.2) **^,#^	−1.6 (0.2) **^,##^
Fat Mass Index (kg/m^2^)	17.1 (6.1)	−2.3 (0.4) **	−3.2 (0.4) **^,##^	−3.3 (0.4) **
Fat-Free Mass Index (kg/m^2^)	21.4 (3.1)	−0.7 (0.2) **	−0.8 (0.2) **	−1.3 (0.2) **^,##^
Skeletal Muscle Index (kg/m^2^)	10.6 (2)	−0.4 (0.2) *	−0.3 (0.2)	−1.3 (0.3) **^,##^
Skeletal Muscle Mass (kg) to VAT (L) ratio	5.4 (0.3)	0.5 (0.3)	0.9 (0.6)	1.8 (0.8) **^,#^
HG (kg)	33.6 (1.6)	0.04 (1.5) *	0.6 (1.2) **	−0.3 (1.5)
MQI (kg/kg)	1 (0.1)	0.2 (0.1) *	0.3 (0.1) **	0.1 (0.1)
Total Body Water (L)	42.8 (9)	−0.2 (0.9)	−0.1 (1)	−0.1 (1.3)
Extracellular Body Water (L)	19.4 (0.6)	−0.1 (0.4)	−0.01 (0.4)	−0.01 (0.6)
ECW to TBW ratio	45.6 (0.4)	−0.1 (0.3)	−0.01 (0.3)	−0.2 (0.3)

Models for each parameter are expressed as mean ± S.E. at T0 and mean variation ± S.E. at T3, T6, and T12. Variation versus T0: * *p* < 0.05; ** *p* < 0.01. Variation versus previous time: # *p* < 0.05; ## *p* < 0.01. Abbreviations: VAT: Visceral Adipose Tissue; HG: Hand-Grip; MQI: Muscle Quality Index; ECW: Extracellular Body Water; TBW: Total Body Water.

## Data Availability

Data supporting the results are available on reasonable request to the corresponding author.

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
