# Peer review of "Once-Weekly Subcutaneous Semaglutide Improves Fatty Liver Disease in Patients with Type 2 Diabetes: A 52-Week Prospective Real-Life Study"

_nutrients, 2022, doi:10.3390/nu14214673_

Round 1

Reviewer 1 Report

This paper focuses on the assessment of the effect of semaglutide on fatty liver disease.

It is refreshingly well written and well presented. The group uses every assessment tool at their disposal to try to measure the presence of NAFLD. Unfortunately, their toolkit is lacking in some of the techniques that can for certain determine the presence of NAFLD and most importantly, of NASH. This is perhaps the major weakness of the study. The presence of NASH with different degrees of fibrosis, cannot be fully confirmed unless higher resolution techniques such as Fibroscan and liver biopsy are used. This differentiation may significantly affect the benefits of semaglutide on liver. However, inferring liver involvement with the techniques used in the Volpe study represents the capabilities of the majority of current clinical institutions and may be the main approach used for a while and, that is why this study still holds value. Thus, I recommend publication with mention of these diagnostic limitations.

1.     The authors stated that ‘to their knowledge, no real-life data assessing the effects of GLP-1 RAs on NAFLD in T2D patients have yet been reported’. However, Newsome et al (PMID: 33185364) reported on a 320 patient study in which they assessed the effect of semaglutide  in patients with  NAFLD/NASH and with/without T2D. This should be stated in the introduction.

The Arai paper (PMID: 35822119) should also be mentioned because, even though it only used 16 patients and oral semaglutide, it evaluated the same patient population using the same parameters.

2.     Both the Newsome and the Arai studies looked into the effect of semaglutide alone. In contrast, Volpe et al. studied the effect of combination therapy (semaglutide/metformin).  This must be mentioned in the introduction and discussion; otherwise it is misleading.

3.      The drug doses of semaglutide used in the Volpe study are much lower than those used in the Newsome study. However, similar benefits on NAFLD were reported by both groups.  One possible explanation is the fact that the Volpe study used metformin in addition to semaglutide, with the drugs probably causing an additive effect. This is important to mention in the discussion.

4.     Ultrasound was the imaging technique used to evaluate NAFLD. The authors do comment in the discussion that this technique is limited in its lack of sensitivity. An important consideration is that it is not possible to know whether the patients in the study, had NAFLD or NASH, a fact that may well affect their response to the drugs. This should be mentioned.

Author Response

  1. The authors stated that ‘to their knowledge, no real-life data assessing the effects of GLP-1 RAs on NAFLD in T2D patients have yet been reported’. However, Newsome et al (PMID: 33185364) reported on a 320 patient study in which they assessed the effect of semaglutide in patients with  NAFLD/NASH and with/without T2D. This should be stated in the introduction.  The Arai paper (PMID: 35822119) should also be mentioned because, even though it only used 16 patients and oral semaglutide, it evaluated the same patient population using the same parameters. 

We thank the Reviewer for the observations. Actually, Newsome et al provided evidence about subcutaneous Semaglutide in NASH, and it was a phase II, randomized, double blinded trial, not a real-life study. This paper was discussed both in the previous and in the present version of the manuscript, in the “Discussion” section. It should be considered that, in the Newsome study, Semaglutide was administered on a daily basis, with a different cumulative exposure compared to our study. To clarify this point, we have added at the end of the “Introduction” section that we evaluated the “effectiveness of diabetes therapeutic dosages of  subcutaneous Semaglutide on fatty liver disease and surrogate markers of NAFLD in patients with T2D, treated once-weekly for 52 weeks”, precisely because it is a real-life study.

The Arai paper is a very interesting paper using a different formulation of Semaglutide, the daily oral one. This has been mentioned in the “Discussion” of this revised version of the manuscript

  1. and 3. Both the Newsome and the Arai studies looked into the effect of semaglutide alone. In contrast, Volpe et al. studied the effect of combination therapy (semaglutide/metformin). This must be mentioned in the introduction and discussion; otherwise it is misleading. 3) The drug doses of Semaglutide used in the Volpe study are much lower than those used in the Newsome study. However, similar benefits on NAFLD were reported by both groups. One possible explanation is the fact that the Volpe study used metformin in addition to Semaglutide, with the drugs probably causing an additive effect. This is important to mention in the discussion. 

This was an important issue that needed clarification, so much so that it was also raised by another reviewer. We think that this observation is very useful to highlight a pharmacological aspect that is often underestimated. Metformin is such a widely used medication in T2D that its discontinuation is usually not required after patient enrollment in RCTs, and the same is observed in clinical practice. This inevitably makes it difficult to discriminate the individual effects of combination therapies and/or to identify potential synergistic effects. In the present version of the manuscript, this issue has been discussed at the end of the “Discussion”, with special mention of the effects of metformin on NAFLD, and a reference has been added  (reference 42).

  1. Ultrasound was the imaging technique used to evaluate NAFLD. The authors do comment in the discussion that this technique is limited in its lack of sensitivity. An important consideration is that it is not possible to know whether the patients in the study, had NAFLD or NASH, a fact that may well affect their response to the drugs. This should be mentioned.

We absolutely agree with this observation, as we have specified in the “limits of the study”. In the “Discussion” section we have added a sentence to express the concept that a “specific investigation in a broader clinical trial…with the aim of clarifying whether the presence of NAFLD or NASH can differently affect the response of patients to Semaglutide” would be highly desirable 

Reviewer 2 Report

Authors Volpe et al. have submitted a manuscript for consideration detailing a 52 week study in T2D patients concerning the effects of Semaglutide add-on therapy with metformin.  The study has some definite strengths that support it, including improvements in metabolic indicators, liver steatosis, BMI, etc.  Given the emergence of GLP-1 RAs as novel therapeutics in combatting T2D and obesity, this study is timely and relevant to the field.  The manuscript is well-written and easy to comprehend. There are some areas that could be improved as follows:

The data, besides the correlation matrix, are limited to tables.  This limits the impact of the findings.  The authors should include graphical representations of the data, even if they are subsets of the indicators used.  Table 4, for instance, could be a supplement, with graphical representations for the most prominent findings and even some that were unaffected by Semaglutide.  

It would be of further interest to include some images from the steatosis determination from the US B-mode images.  This could inform readers of the importance of this technique besides biopsy.  

One lacking aspect of this study the authors should comment upon is that this study is, in effect, describing a combination therapy.  This does not detract from the findings, but should have consideration in the discussion.  That is to say since patients are all on metformin, the Semaglutide effect is potentially a synergistic mechanism.  The authors should look into what possibly drives this combinatorial effect in the literature and add to discussion, as it is briefly mentioned only once in the Study Protocol.  

Author Response

  • The data, besides the correlation matrix, are limited to tables.  This limits the impact of the findings.  The authors should include graphical representations of the data, even if they are subsets of the indicators used.  Table 4, for instance, could be a supplement, with graphical representations for the most prominent findings and even some that were unaffected by Semaglutide.

We agree with the reviewer’s suggestion, and have provided graphical implementation, as you can see in the present version of the manuscript. More precisely, Table 4 was reorganized to include only information on body composition data (bio-impedance parameters). All other results, previously described in table 4, have been graphically represented (Figures from number 4 to number 8). As suggested, results were also shown for those parameters that did not significantly change due to treatment with Semaglutide. At the same time, Figures 2 and 3 have been added to respond to the request of another reviewer.

  • It would be of further interest to include some images from the steatosis determination from the US B-mode images. This could inform readers of the importance of this technique besides biopsy.

We are particularly grateful for this request. We were initially concerned about a possible redundant effect of showing US images. According to your suggestion, we have gladly added US images in “Figure 1”, with explicative purposes.

  • One lacking aspect of this study the authors should comment upon is that this study is, in effect, describing a combination therapy. This does not detract from the findings, but should have consideration in the discussion.  That is to say since patients are all on metformin, the Semaglutide effect is potentially a synergistic mechanism.  The authors should look into what possibly drives this combinatorial effect in the literature and add to discussion, as it is briefly mentioned only once in the Study Protocol.

This observation is very useful to highlight a pharmacological aspect that is often underestimated. Metformin is such a widely used medication in T2D that its discontinuation is usually not required after patient enrollment in RCTs, and the same is observed in clinical practice. This inevitably makes it difficult to discriminate the individual effects of combination therapies and/or to identify potential synergistic effects. In the present version of the manuscript, this issue has been discussed at the end of the “Discussion” with special mention of the effects of metformin on NAFLD, and a reference has been added  (reference 42).

Reviewer 3 Report

The manuscript is interesting, but your way of organizing the ideas is distract the reader. In my opinion, the number, age, sex and grouping of participants in the experiment should be described first in the material and method, and then the specific method of the experiment and the observed indicators should be described. Besides, I have something to know

(1)The article shows that 150 people participated in the clinical study and 48 people received drug treatment. Why didn't the author compare the observed indicators of 48 people who participated in drug treatment with 102 people who did not receive drug treatment?

(2)Are there gender specific observations in the study?

(3)lin 231-232 : The patients were predominantly men (54.2%), with a male to female ratio (1.2:1) should mentioned in the methods firstly.

(4)Baseline characteristics of study population based on 48 people or 150 ?

(5)the results shown in line 232-252  could not be seen in Table 2,

(6)in the part of Body composition, make a comparative analysis according to gender, whether there are differences in the data related to body composition of drug therapy in terms of gender, so as to obtain more information

(7)line 338-346 Are the results of the author's research consistent with those of the predecessors? What are the differences? Why?

(8)reorganize the discussion, which seems to be confusing. The author can first introduce the research results of this paper by citing the previous research results, and then analyze the possible reasons for the consistency or difference with the previous research.

(9)line 365-366 Why a separate paragraph? You'd better to merge with other paragraphs

(10)line 376-377:。。。。lastly preventing or 376 halting atherosclerosis, add reference

Author Response

The manuscript is interesting, but your way of organizing the ideas is distract the reader. In my opinion, the number, age, sex and grouping of participants in the experiment should be described first in the material and method, and then the specific method of the experiment and the observed indicators should be described.

As requested, the description of the number, age, sex and grouping of patients, contained in the paragraph “Screening for eligibility of study participants” (2.6 in the previous version of the manuscript) has been moved up. Please see paragraph 2.2 in the “Methods” section.

  • The article shows that 150 people participated in the clinical study and 48 people received drug treatment. Why didn't the author compare the observed indicators of 48 people who participated in drug treatment with 102 people who did not receive drug treatment?

The study design was to specifically evaluate the effects of once-weekly Semaglutide on NAFLD. Since the remaining 102 patients were largely treated with other antidiabetic drugs (SGLT2i and other GLP1-RAs), a new study is already underway, to compare the effects of these drugs on the observed indicators.

  • Are there gender specific observations in the study? 6) In the part of Body composition, make a comparative analysis according to gender, whether there are differences in the data related to body composition of drug therapy in terms of gender, so as to obtain more information

Thank you for these observations. Actually, in the previous version of the manuscript, we only specified that no gender differences emerged from the study regarding the effects of Semaglutide on NAFLD, which was the primary study outcome (see paragraph 3.3 in the “Results” section).

However, the assessment had been performed in the study population as a whole and separately for men and women, and we found that changes in variables during follow-up were comparable between genders. This clarification has been provided in the revised version of the manuscript, both in paragraph “2.7 Statistical Analysis" of the “Methods” section, and at the end of paragraph “3.4 Body composition” of the “Results” section.

  • Lin 231-232: The patients were predominantly men (54.2%), with a male to female ratio (1.2:1) should mentioned in the methods firstly.

As required, the description has been mentioned in paragraph 2.2.

  • Baseline characteristics of study population based on 48 people or 150?

Baseline characteristics are referred to patients treated with Semaglutide. To clarify this point this concept has been specified in the Table 2 title (n = 48).

  • The results shown in line 232-252 could not be seen in Table 2.

We are sorry to inform you that we did not find a precise matching with lines indicated by the reviewer. We assume that the reviewer referred to the entire paragraph 3.1. To provide fruitful changes to the manuscript, two figures (Figure 1 and Figure 2) have been added to illustrate results previously described in the text only.

In the third column are summarized median values with their related ranges. Some remarks on the results refer to the third column. To provide more detailed results, some numerical values ​​have also been reported in the text.

  • See point number 2

  • In line 338-346 Are the results of the author's research consistent with those of the predecessors? What are the differences? Why.

As mentioned above, we are not sure that we correctly identified the line numbering recalled by the reviewer. We assume that the reviewer was referring to data from Newsome 2019-2021 and Flint 2021 (References 38, 40 and 41 of the previous version of the manuscript). In the present version, we specified that our real-life data are in line with these clinical trials. However, it should be noted that, in our study, we cannot go so far as to speculate on liver fibrosis, as like in the Newsome and Flint studies,  because we lack fibroscan data.

  • Reorganize the discussion, which seems to be confusing. The author can first introduce the research results of this paper by citing the previous research results, and then analyze the possible reasons for the consistency or difference with the previous research.

The discussion has been reorganized as suggested. First, the research findings are introduced and discussed. The data from the literature were consistent with ours, and this concept has been pointed out at the end of the manuscript.

  • Line 365-366 Why a separate paragraph? You'd better to merge with other paragraphs

The paragraph has been merged with the others.

  • Line 376-377:。。。。lastly preventing or 376 halting atherosclerosis.. add reference

The required reference has been added

Round 2

Reviewer 3 Report

The author then modifies the format according to the magazine requirements . recommends publication